# Controllability of Graphene Oxide Doxorubicin Loading Capacity Based on Density Functional Theory

**DOI:** 10.3390/nano12030479

**Published:** 2022-01-29

**Authors:** Jiaming Song, Naiyu Cui, Shixun Sun, Xinyue Lu, Yuxuan Wang, Haoyu Shi, Eui-Seok Lee, Heng-Bo Jiang

**Affiliations:** 1The CONVERSATIONALIST Club, School of Stomatology, Shandong First Medical University and Shandong Academy of Medical Sciences, Tai’an 271016, China; jiamingsongpp@outlook.com (J.S.); cuinaiyuuu@outlook.com (N.C.); sunshixx@gmail.com (S.S.); xinyuelu111@outlook.com (X.L.); wangaxuan2020@outlook.com (Y.W.); mtdykxy@hotmail.com (H.S.); 2Department of Oral and Maxillofacial Surgery, Graduate School of Clinical Dentistry, Korea University, Seoul 08308, Korea

**Keywords:** graphene, doxorubicin, density functional theory, drug delivery, reduced density gradient

## Abstract

Graphene can be used as a drug carrier of doxorubicin (DOX) to reduce the side effects of doxorubicin. However, there is limited research on the surface chemical modifications and biological effects of graphene oxide (GO). Therefore, it is necessary to explore the DOX affinity of different oxygen-containing functional groups in the graphene system. We constructed graphene system models and studied the structure and distribution of epoxy and hydroxyl groups on the carbon surface. Based on molecular dynamics simulations and density functional theory (DFT), we investigated the interaction between DOX and either pristine graphene or GO with different ratios of oxygen-containing groups. The hydroxyl groups exhibited a stronger affinity for DOX than the epoxy groups. Therefore, the DOX loading capacity of graphene systems can be adjusted by increasing the ratio of hydroxyl to epoxy groups on the carbon surface.

## 1. Introduction

Graphene and its derivatives have great research value and application prospects in various fields because of their unique structures [1,2]. In particular, graphene is increasingly being used in biomedicine because of its excellent drug delivery properties [3,4]. However, the high hydrophobicity of raw graphene prevents solubility and dispersion, and the cytotoxicity makes hemolysis likely, limiting drug delivery applications [5,6]. A derivative of graphene, graphene oxide (GO), has a large number of hydrophilic groups, such as hydroxyl and carboxyl groups, on the surface and edges, which confer stability and wettability in aqueous solutions and polar solvents. Furthermore, GO disperses well in water, an essential property of drug carriers [7,8]. GO can be modified to increase its stability and safety and is suitable for clinical applications [9,10,11]. Dai et al. demonstrated that functionalized nanographene tablets with good biocompatibility and no significant cytotoxicity can be loaded with aromatic anticancer drugs. Yang et al. demonstrated that GO can be a highly effective aromatic anticancer drug carrier by preparing a novel GO-doxorubicin (DOX) hydrochloride nanohybrid material [12,13]. However, GO-based nanoscale drug carriers have not been fully studied, and it is necessary to improve the drug loading efficiency of GO through functionalization and surface modification.

DOX, a common clinical anticancer drug, inhibits the DNA replication of cancer cells [14]. DOX has remarkable therapeutic effects on various tumors [15,16,17]. However, because of its strong cytotoxicity and damaging effects on normal cells, it is often used clinically in combination with drug carriers [18].

GO contains both sp3- and sp2-hybridized carbon atoms, with the sp3-hybridized carbon atoms mainly involved in oxygen-containing functional groups. Therefore, different interaction mechanisms are involved in the adsorption process between DOX and GO. Tonel et al. reported that the most stable conformation is achieved when DOX is parallel to pristine graphene [2]. Vovusha et al. showed that adsorption between DOX and the sp2 region of GO is stronger than that at the sp3 region [5]. Therefore, in order to use graphene systems as drug carriers, specialized surface and functional modifications are essential. These prior studies have provided a theoretical basis for exploring the interactions between DOX and carbon nanostructures.

Understanding the interaction between the graphene system and DOX, especially the role of the functional groups in the adsorption process, is key. A previous study confirmed that functional groups play important roles in the adsorption of DOX to carbon nanotube drug carriers [14,19]. The hydrogen bonds between DOX and the hydroxyl groups of the carbon nanotubes significantly affect the adsorption and immobilization processes [9]. It has been reported that variations in surface chemistry may result in changes in biochemical toxicity and solubility [20]. Density functional theory (DFT) can be used to understand the adsorption mechanism of anticancer drugs and inform the regulation of the ratio of oxygen-containing functional groups. Different oxygen-containing functional groups in GO have different affinities for DOX; therefore, the loading of anticancer drugs can be regulated by modifying these groups. Understanding the interaction between oxygen-containing groups and anticancer drugs is essential for realizing the clinical applications of GO as a drug carrier.

Research on molecular interactions and the ratio of epoxy to hydroxyl groups on the surface of GO is limited. Therefore, we constructed graphene system models and studied the structure and distribution of epoxy and hydroxyl groups on the carbon surface. DFT was used to investigate the mechanism of the interaction between DOX and pristine graphene, GO with different ratios of oxygen-containing groups. DFT and charge density difference analyses were also conducted. By exploring the affinities between DOX and different oxygen-containing functional groups and adjusting the ratio of epoxy to hydroxyl groups, we aimed to provide a theoretical basis for increasing the DOX loading capacity of graphene systems.

## 2. Materials and Methods

### 2.1. Model Building

#### 2.1.1. DOX

The chemical structure of DOX is shown in Figure 1. DOX belongs to the anthracycline class. It has two parts: a four-ring aromatic hydroxyanthraquinone ring and an aminoglycoside ring. DOX contains functional groups such as carbonyl, hydroxyl, and amino groups.

#### 2.1.2. Pristine Graphene

The pristine graphene surface is composed of 308 carbon atoms, with a surface area of 25.926 Å × 28.495 Å, and the length of the C-C bonds is 1.42 Å (Figure 2). The surface was modeled using periodic boundary conditions and A vacuum plate of 35 Å thickness was added to avoid atomic interactions between adjacent layers.

#### 2.1.3. GO

Due to the non-stoichiometric chemical composition of GO, it is difficult to establish a GO model. The GO surface contains many hydroxyl and epoxy groups, while the edges contain a small number of carbonyl and carboxyl groups; therefore, this study only considered C_8_O_2_(OH)_2_ units containing epoxy and hydroxyl groups [21]. C_8_O_2_(OH)_2_ is considered a stable unit of fully oxidized GO structures [22]. However, the exact arrangement of epoxy groups and hydroxyl groups in C_8_O_2_(OH)_2_ has not been reported. The two hydroxyl groups are above the carbon plane; the epoxy groups can either both be below the carbon plane or opposite each other (one on each side of the plane) [23,24].

Here, two different C_8_O_2_(OH)_2_ unit structures were established. In one, both epoxy groups are below the carbon plane, and both hydroxyl groups are above the carbon plane (Figure 3a,b). In this unit, by adsorbing DOX onto both sides, the difference in the affinity of the hydroxyl and epoxy groups for DOX was explored. In the other structure, both hydroxyl groups were above the carbon plane, while the epoxy groups were opposite each other (Figure 3c). In order to achieve sufficient interaction between DOX and C_8_O_2_(OH)_2_ units, the GO model used in this study contained three C_8_O_2_(OH)_2_ units, oriented to form an equilateral triangle.

### 2.2. Molecular Dynamics Simulation

To establish a more accurate adsorption model, we use molecular dynamics simulations to simulate the adsorption trajectory and search for rough global optimal adsorption structures, then use this structure as a starting point for DFT optimization. All DFT calculations were performed using CP2K. The Nosé thermostat was used to maintain the temperature at 300 K. A constant volume was maintained, and 2000 steps were calculated for sampling.

### 2.3. Structure Optimization

The structure with the least energy is selected from the molecular dynamics simulation for optimization. DFT calculations were based on a mixed Gaussian and plane wave (GPW) approach, Perdew-Burke-Ernzerhof (PBE) exchange-related functionals, and the corresponding pseudo potentials [25,26,27]. A 500 Ry plane wave density cutoff and periodic boundary conditions were used [28]. Empirical dispersion corrections were implemented using the Grimme D3 method [29]. The stopping criterion for geometric optimization and energy calculation are set as follows: (a) the maximum interatomic force is less than 4.5 × 10^−4^ Bohr/Hartree; (b) the force between root mean square atoms is less than 3 × 10^−4^ Bohr/Hartree; (c) the maximum displacement is less than 3 × 10^−3^ Bohr; (d) the root mean square atomic displacement is less than 1.5 × 10^−3^ Bohr.

### 2.4. Adsorption Energy

The strength of DOX adsorption to the graphene surface can be directly reflected by the adsorption energy, which is obtained according to the following equation:(1)$$E_{\mathrm{ads}}=E_{\mathrm{GS}+\mathrm{DOX}}-\left(E_{\mathrm{GS}}+E_{\mathrm{DOX}}\right)$$
where *E*_GS+DOX_, *E*_GS_, and *E*_DOX_ represent the energies of the complex, graphene, and DOX, respectively. A negative value of *E*_ads_ indicates that the adsorption system is stable. 

### 2.5. Reduced Density Gradient (RDG) Analysis

RDG analysis is an extremely useful weak interaction analysis method [30]. It can be performed using the Multiwfn program for electronic wavefunction analysis [31]. Only the calculated wave function file needs to be entered to analyze and visualize multiple types of noncovalent interactions in real space. The RDG can be calculated using the following equation:(2)$$\mathrm{RDG}(r)=\frac{1}{2{\left(3\pi^{2}\right)}^{1/3}}\;\frac{\left|\nabla \rho \left(r\right)\right|}{\rho {\left(r\right)}^{4/3}}\rho \left(r\right),$$
where *ρ*(*r*) represents electron density, and |∇*ρ*(*r*)| represents the norm of the electron density gradient vector.

### 2.6. Charge Density Difference Analysis

To visually observe the change in electron density when DOX is adsorbed onto the graphene system surface, a charge density difference analysis was conducted, using the following equation:(3)$$\Delta \rho =\rho_{\mathrm{GS}+\mathrm{DOX}}-\rho_{\mathrm{GS}}-\rho_{\mathrm{DOX}}$$
where $\rho_{\mathrm{GS}+\mathrm{DOX}}$, $\rho_{\mathrm{GS}},\;\mathrm{and}\;\rho_{\mathrm{DOX}}$ represent the electron densities of the complex, graphene system, and DOX, respectively.

## 3. Results and Discussion

### 3.1. Structure and Energy Analysis

We selected a number of relatively stable adsorption structures from the structures sampled using the molecular dynamics simulation and then optimized them to obtain the most stable structure for analysis. Table 1 shows the adsorption energies of the stable structures. The adsorption energy was negative, so adsorption between DOX and the graphene system was thermodynamically favorable.

In this study, the adsorption energies of the DOX and graphene systems were slightly higher than those observed in a previous study. By using a molecular dynamics simulation as the sampling method, we found that the orientation of DOX changed so that the anthraquinone and aminoglycoside rings faced the graphene. Thus, DOX had more sites of interaction with the carbon surface, which is the main reason for the higher adsorption energies observed. Conversely, in the study by Vovusha et al., only one of the two rings faced the graphene [5]. The adsorption of GO surface and doxorubicin molecules were also affected by water in the physiological environment, which may compete with DOX for the active sites on GO surface and to some extent weaken the interaction between GO and DOX. As a result, the binding strength between GO and DOX in the physiological environment is weaker than the result obtained by our calculation which were performed in a vacuum.

The maximum adsorption energy was observed between DOX and pristine graphene, which indicates that the adsorption of DOX onto pristine graphene was the most favorable (Figure 4). The π-conjugated structure of graphene formed a π–π stacking interaction with the quinone portion of DOX, producing a hydrophobic effect. The distance between graphene and DOX was about 3.2 Å. CH-π and OH-π interactions between DOX and pristine graphene were also observed.

In the GO-OH-DOX complex (Figure 5), the two carbonyl groups in the anthraquinone ring of DOX formed hydrogen bonds with the hydroxyl groups and hydrogen atoms on the GO-OH surface. The bond lengths were, respectively, 2.05 Å and 2.21 Å. The -O- of DOX also formed a hydrogen bond (1.87 Å) with the hydrogen in the hydroxyl group. In the GO-O-DOX complex (Figure 6), there were only a few weak hydrogen bonds; the bond between a carbonyl group in the anthraquinone ring of DOX and an H atom on the GO surface was 2.05 Å long. The aminoglycoside ring of DOX also formed CH-π and OH-π interactions with the sp2 region of the GO surface. Two hydrogen bonds were formed between DOX and GO in the GO-OH-O-DOX complex (Figure 7). The length of the bond between a carbonyl group in the anthraquinone ring on DOX and an H atom on the GO surface was 2.01 Å long. The H atom in the hydroxyl group of DOX formed a hydrogen bond of 2.62 Å with the oxygen in the hydroxyl group on the GO surface. NH2-π and OH-π interactions between the aminoglycoside ring of DOX and the sp2 region of the GO surface were also observed.

The interactions between DOX and the graphene system can have a significant effect on the hydrogen bonds within the DOX molecules. Table 2 shows the lengths of the hydrogen bonds within the DOX and graphene system molecules. The hydrogen bonds within the DOX molecules were slightly shortened, and when DOX was adsorbed onto the GO-OH-O- surface, the O4-H bond broke, and an N-H bond was formed.

The adsorption energy between GO and DOX was smaller than that between pristine graphene and DOX. This was mainly because of the strong π–π stacking interaction between pristine graphene and DOX. Our results were consistent with previous studies of the interaction between aromatic compounds and pristine graphene [32,33]. However, due to the introduction of oxygen-containing functional groups, some of the sp2-hybridized carbon atoms become sp3 hybrids. Therefore, the π-π conjugated structure of GO was weakened, as was the π-π stacking effect between DOX and GO. The adsorption energy of the GO-DOX complex was greater than that of the GO-O-DOX complex. The affinity for DOX differed between the hydroxyl and epoxy groups, possibly because the electronegative oxygen atoms of DOX more easily formed hydrogen bonds with the highly *Lewis* acidic hydrogen atoms on the hydroxyl-rich GO surface. Conversely, the epoxy-rich GO surface provided fewer acidic hydrogen atoms and, subsequently, formed weaker hydrogen bonds, resulting in less affinity for DOX.

### 3.2. RDG Analysis

The CH-π and OH-π interactions between the anthraquinone ring in DOX and pristine graphene can be clearly observed in the color-filled RDG isosurface plots (Figure 8). The intermittent green fragment between DOX and GO corresponds to a weak π-π stacking interaction between DOX and GO [13]. When comparing the GO-OH-DOX, GO-O-DOX, and GO-OH-O-DOX complexes, the green fragments in GO-O-DOX are relatively complete and have a large area, indicating that the π−π stacking interaction between the GO-O surface and DOX was stronger in GO-O-DOX than in the other complexes. However, the hydrogen bond between the GO-O surface and DOX was weaker in the GO-O-DOX complex, and the adsorption energy was the lowest. Therefore, it can be concluded that the adsorption between GO and DOX was mainly due to hydrogen bonding.

### 3.3. Charge Density Difference

To observe the charge transfer between graphene and DOX, we performed a charge density difference analysis (Figure 9). There was a large charge transfer between the graphene system surface and the DOX anthraquinone ring; a charge transfer also occurred at other sites where DOX interacted with the graphene system. We also performed a Bader charge analysis and found that when DOX was adsorbed onto pristine graphene, the pristine graphene transferred 0.04 electrons to DOX. When DOX was adsorbed onto GO, 0.05 electrons were transferred from DOX to GO-O and GO-OH-O surface, respectively. Additionally, 0.07 electrons are transferred from DOX to the GO-OH surface.

## 4. Conclusions

Using DFT, we analyzed the interaction between DOX and either pristine graphene or GO loaded with different ratios of oxygen-containing functional groups. We observed that different oxygen-containing functional groups had different affinities for DOX. The order of the different graphene systems was as follows, in terms of adsorption energy: G-DOX > GO-OH-DOX > GO-OH-O-DOX > GO-O-DOX. Therefore, increasing the ratio of hydroxyl to epoxy groups can increase the DOX loading capacity of GO. This provides a theoretical basis for optimizing the surfaces of graphene substrate materials used to load DOX.

## Figures and Tables

**Figure 1 nanomaterials-12-00479-f001:**
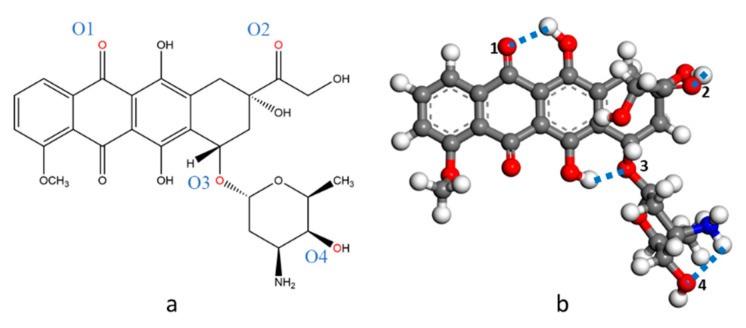
Representation of doxorubicin (DOX) structure. (**a**) 2D; (**b**) 3D. Color code: C, gray; O, red; N, blue; H, white.

**Figure 2 nanomaterials-12-00479-f002:**
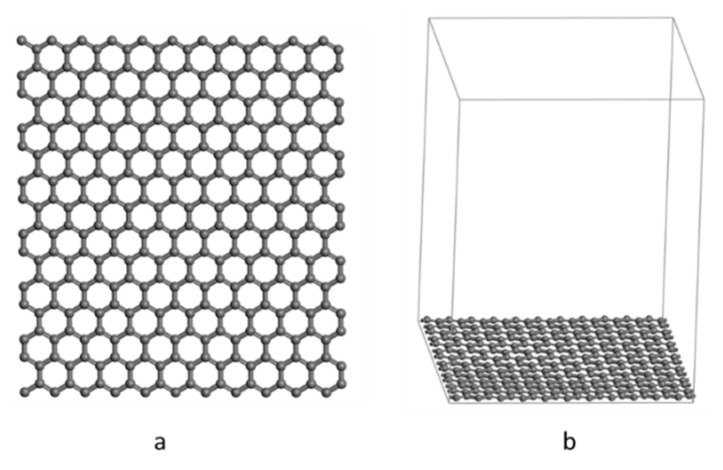
Representation of pristine graphene surface. (**a**) Graphite surface structure; (**b**) graphene structure with vacuum layer added.

**Figure 3 nanomaterials-12-00479-f003:**
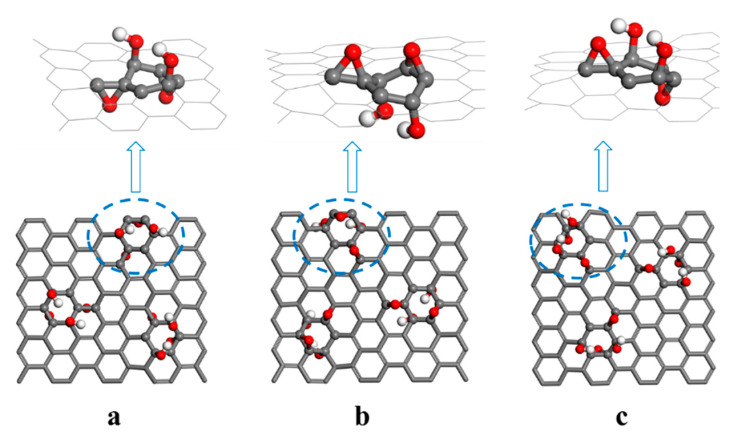
(**a**–**c**) Representation of three graphene oxide (GO) models.

**Figure 4 nanomaterials-12-00479-f004:**
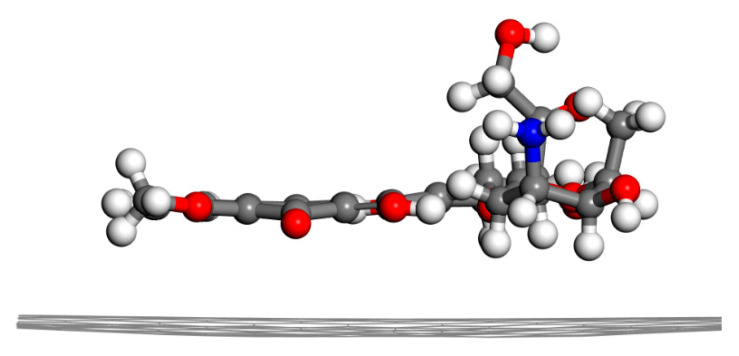
Energy-minimized structure of G-DOX (side view).

**Figure 5 nanomaterials-12-00479-f005:**
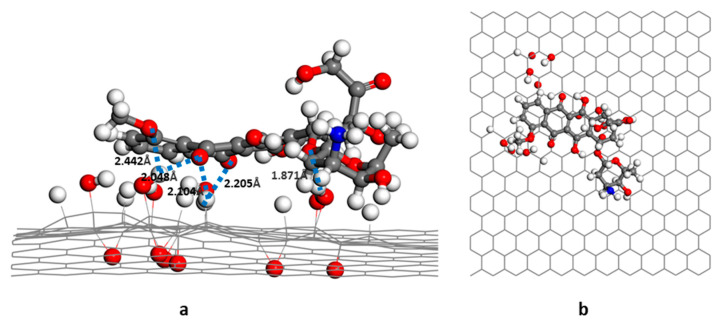
Energy-minimized structure of GO-OH-DOX; (**a**) side view; (**b**) top view.

**Figure 6 nanomaterials-12-00479-f006:**
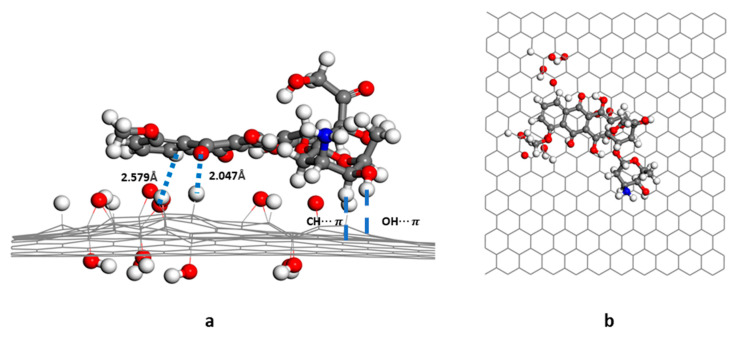
Energy-minimized structure of GO-O-DOX; (**a**) side view; (**b**) top view.

**Figure 7 nanomaterials-12-00479-f007:**
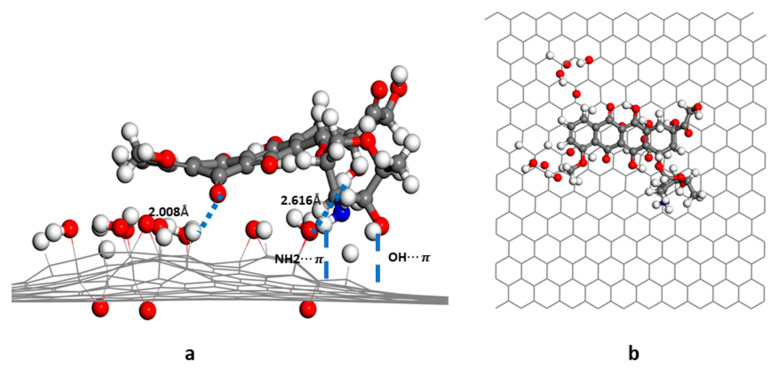
Energy-minimized structure of GO-OH-O-DOX; (**a**) side view; (**b**) top view.

**Figure 8 nanomaterials-12-00479-f008:**
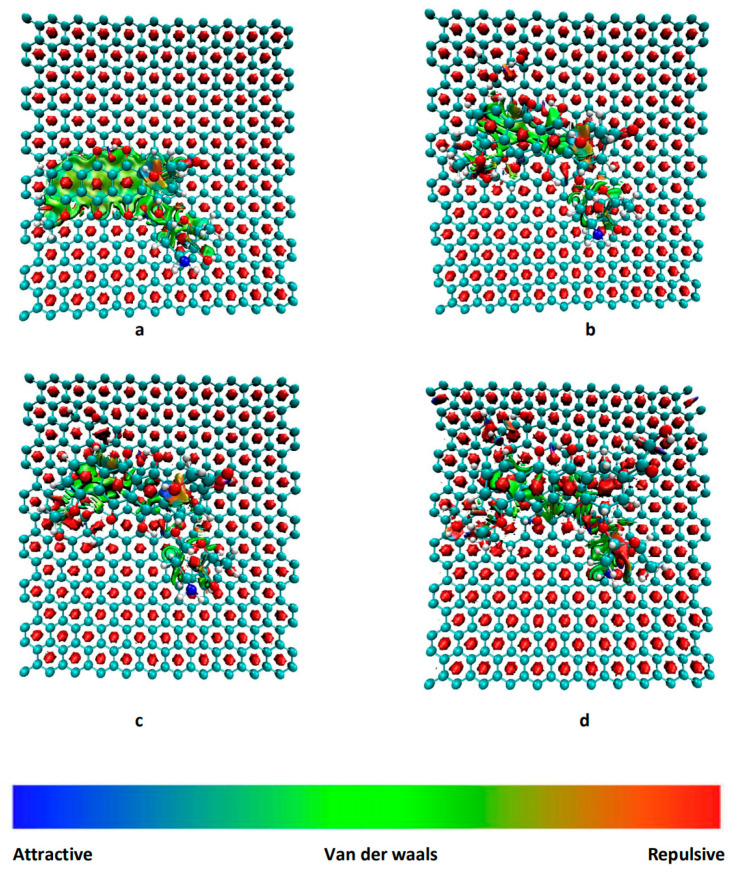
Reduced density gradient isosurface of different molecular graphene surfaces. (**a**) G-DOX; (**b**) GO-OH-DOX; (**c**) GO-O-DOX; (**d**) GO-OH-O-DOX.

**Figure 9 nanomaterials-12-00479-f009:**
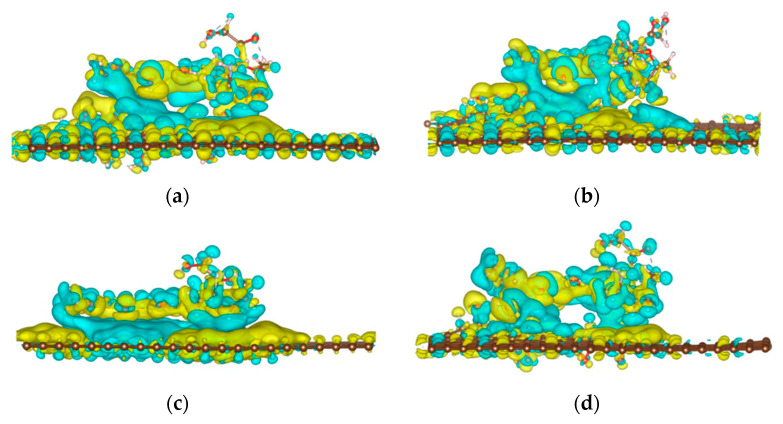
Differential charge density of different molecular graphene surfaces. (**a**) G-DOX; (**b**) GO-OH-DOX; (**c**) GO-O-DOX; (**d**) GO-OH-O-DOX. The yellow and blue areas represent an increase and decrease in charge density, respectively.

**Table 1 nanomaterials-12-00479-t001:** Adsorption energy of the stable structure (eV).

Models	$E_{GS}+E_{DOX}$	$E_{GS+DOX}$	$E_{GS}$	$E_{ads}$
G-DOX *	−57,483.232	−57,480.219	−47,764.498	−3.013
GO-OH-DOX	−62,918.418	−62,915.918	−53,199.578	−2.501
GO-O-DOX	−62,918.477	−62,915.652	−53,199.578	−2.825
GO-OH-O-DOX	−62,916.538	−62,913.910	−53,198.480	−2.628

* G, graphene.

**Table 2 nanomaterials-12-00479-t002:** Hydrogen bond length of DOX and graphene system molecules (Å). The annotation of O1 to O4 have shown in Figure 1.

Models	O_1_·········· ·H	O_2_··· ···H	O_3_··· ···H	O_4_······ ···H	N··· ···H
DOX	1.812	1.986	1.738	2.305	-
G-DOX	1.501	1.906	1.648	2.426	-
GO-OH-DOX	1.598	1.869	1.753	2.370	-
GO-O-DOX	1.515	1.826	1.682	2.373	-
GO-OH-O-DOX	1.551	2.039	1.626	-	1.903

## Data Availability

Not applicable.

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
