# Peer review of "Controllability of Graphene Oxide Doxorubicin Loading Capacity Based on Density Functional Theory"

_nanomaterials, 2022, doi:10.3390/nano12030479_

Round 1
Reviewer 1 Report
The text under review is a purely computational study of adsorption of widely used anticancer agent molecules (doxorubixin - DOX) on the surface of graphene (G) and graphene oxide (GO). The last contains functionalization groups of two types hydroxyl (GO-OH) and epoxy (GO-O).
Authors applied the density functional theory on the PBE level with dispersive correction to obtain the structures with optimal total energy. They calculated adsorption energies and performed topological analysis of the non-bonded interactions between atoms.
In the result of the consideration of adsorption energies the most favorable is the unmodified graphene substrate. Next favorable is the graphene oxide with hydroxyl groups, and the least – graphene oxide with epoxide groups.
Also authors found that hydrogen bonds are more important than π-π (van der Waals?) interactions in the DOX adhesion process on the GO.
Broad/major comments:
1. Though the results are interesting they, up to my opinion, does not show significant progress comparing to the published results, which were cited by the authors. May be the more thorough comparison with literature will stress the importance of the presented study. In particular, the reference 5 of the text cites the paper with title covering this study “Binding Characteristics of Anticancer Drug Doxorubicin with 2D Graphene and Graphene Oxide: Insights from Density Functional Theory Calculations and Fluorescence Spectroscopy”. The progress comparing to this study must be explicitly mentioned.
2. The conclusions looks very weak. Do we really need to perform DFT calculations to found that “increasing the ratio of hydroxyl to epoxy groups can increase the DOX loading capacity of GO.” Especially if unmodified graphene already provides better adhesion? No quantitative values of “loading capacity” is presented, while this could be expected because of the title.
Specific/minor comments:
3. GO model structure. Authors considered 3 variants of C8O2(OH)2 fragment of GO. Could be considered the 4th variant, with OH and O groups above surface, and other OH and O group below surface? I suspect that such configuration will have saturated hydrogen bonds, and thus will be the most stable and probable. Also, the DFT calculations should allow to select the most probable GO configuration among the considered, why it was not done?
4. Some details of the calculation methods are missing. Why 300 K was selected for MD simulations? Is it enough to obtain all structural configurations? What was the stopping criterion for structural optimization? Why PBE functional was selected? May be B3LYP is more suitable for considered systems? Why 500 Ry cutoff was used? Were there performed convergence tests? Why Grimme D3 method for dispersive corrections was selected? In ref. 5 was used another method.
5. Inconsistencies in the text. Equation (1) is abrupt. RDG analysis uses wave-functions according to Eq.(2), but authors claims that “only the atomic coordinates need to be entered”. Abbreviations in Table 1 are used before introduction in the text. Notations O1, O2, O3 and O4 used in Table 2 are not explained.
6. Values of the charge transfer are desirable in the Section 3.3. I suspect that absolute charge transfer is negligible.
7. Suggestion. To achieve significant progress comparing to the study [5] I would suggest to consider the borders of G and GO, at least zigzag ones. The target biological application makes highly desirable the consideration of the solvent molecules, i.e. water. How extra hydrogen bonds with H2O molecules could alter the DOX adhesion?
Author Response
Broad/major comments:
Q1: Though the results are interesting they, up to my opinion, does not show significant progress comparing to the published results, which were cited by the authors. May be the more thorough comparison with literature will stress the importance of the presented study. In particular, the reference 5 of the text cites the paper with title covering this study “Binding Characteristics of Anticancer Drug Doxorubicin with 2D Graphene and Graphene Oxide: Insights from Density Functional Theory Calculations and Fluorescence Spectroscopy”. The progress comparing to this study must be explicitly mentioned.
Response: Reference 5 mainly compared the adsorption differences of doxorubicin on graphene oxide and pristine graphene, and proved that pristine graphene has a stronger binding ability to doxorubicin. Our work focused on analyzing the difference in the binding ability of doxorubicin with go containing different proportions of oxygen functional groups, in order to demonstrate the possibility of changing the ratio of oxygen functional groups on the surface of graphene oxide to change doxorubicin loading. And because we used molecular dynamics simulations to find the most stable structure of the adsorption complex. Therefore, we found a better adsorption load than reference 5, which can be seen from the slightly higher adsorption energy calculated by us.
Q2: The conclusions looks very weak. Do we really need to perform DFT calculations to found that “increasing the ratio of hydroxyl to epoxy groups can increase the DOX loading capacity of GO.” Especially if unmodified graphene already provides better adhesion? No quantitative values of “loading capacity” is presented, while this could be expected because of the title.
Response: Although unmodified graphene has provided better adhesion, raw graphene's high hydrophobicity prevents dispersion in the blood and the formation of hemolysis, thus limiting its use as a drug carrier. Go has higher stability and safety, and is more suitable as a drug carrier. However, doxorubicin has weak binding ability to GO, so it is very necessary to find a method to improve the binding ability of GO and doxorubicin. The DFT study is very convenient and clear. We can simulate the interaction of doxorubicin with different ratios of hydroxyl and epoxy groups by adjusting the go modeling. Not only can the ratio of different hydroxyl and epoxy groups and the difference of doxorubicin adsorption intensity, but also can analyze the different adsorption mechanism. However, DFT can only be used to calculate a simplified model, which cannot simulate the influence of other factors in the real environment. In the future, we will conduct experiments based on this calculation result to verify the specific influence of the ratio of different hydroxyl and epoxy groups on the binding strength of doxorubicin in the biological environment.
Specific/minor comments:
Q3: GO model structure. Authors considered 3 variants of C8O2(OH)2 fragment of GO. Could be considered the 4th variant, with OH and O groups above surface, and other OH and O group below surface? I suspect that such configuration will have saturated hydrogen bonds, and thus will be the most stable and probable. Also, the DFT calculations should allow to select the most probable GO configuration among the considered, why it was not done?
Response: Because according to Ghaderi et al., two hydroxyl groups in C8O2(OH)2 unit have the lowest energy when they are on the same side of the carbon plane. So we're not thinking about the oh on either side of the carbon plane.
Ghaderi, N.; Peressi, M. First-Principle Study of Hydroxyl Functional Groups on Pristine, Defected Graphene, and Graphene Epoxide. J. Phys. Chem. C 2010, 114 (49), 21625–21630. doi:10.1021/jp108688m.
Q4: Some details of the calculation methods are missing. Why 300 K was selected for MD simulations? Is it enough to obtain all structural configurations? What was the stopping criterion for structural optimization? Why PBE functional was selected? May be B3LYP is more suitable for considered systems? Why 500 Ry cutoff was used? Were there performed convergence tests? Why Grimme D3 method for dispersive corrections was selected? In ref. 5 was used another method.
Response: Thanks for your suggestion. We have added more calculation details in the manuscript. Simulation at 300K May better reflect the temperature in the real environment. The purpose of MD is to find the adsorption structure with the minimum energy, that is, the most stable adsorption structure. Therefore, we only select the structure with the minimum energy from the structure obtained by MD for subsequent DFT structure optimization. The stopping criterion of structural optimization is that the maximum interatomic force is less than 4.5E-4 Bohr/Hartree, the interatomic force is less than 3E-4 Bohr/Hartree, the maximum displacement is less than 3E-3 Bohr, and the atomic displacement is less than 1.5E-3 Bohr. Since our system is a large periodic system, B3LYP is weaker than PBE functional in a larger atomic number and periodic system, and graphene is a large conjugated system. B3LYP's calculation results for a large range of conjugated systems are inferior to that of PBE functional. The truncation value of 500 Ry was obtained by the convergence test. Grimme D3 is an updated version of DFT-D2 with better overall accuracy than DFT-D2. It hardly adds any time to the computation.
Q5: Inconsistencies in the text. Equation (1) is abrupt. RDG analysis uses wave-functions according to Eq.(2), but authors claims that “only the atomic coordinates need to be entered”. Abbreviations in Table 1 are used before introduction in the text. Notations O1, O2, O3 and O4 used in Table 2 are not explained.
Response: Due to our typesetting mistake, Equation (1) has been omitted. We have added the Equation (1) in the revised manuscript. The calculation of RDG is obtained by wave function. What we want to express in this paper is that Wave function analysis software Multiwfn can either directly calculate RDG using atomic coordinates or wave functions. Thank you for reminding us that we have corrected this inaccurate expression in the manuscript. For the symbols O1, O2, O3 and O4 used in Table 2, annotations are added in Table 2 to make readers understand more clearly in Figure 1, and the classical structure of DOX is added in Figure 1 so that the atoms corresponding to symbols O1, O2, O3 and O4 can be clearly understood.
Q6: Values of the charge transfer are desirable in the Section 3.3. I suspect that absolute charge transfer is negligible.
Response: Yes. although there is a partial charge transfer between DOX and the adsorption surface the absolute value is very small.
Q7: Suggestion. To achieve significant progress comparing to the study [5] I would suggest to consider the borders of G and GO, at least zigzag ones. The target biological application makes highly desirable the consideration of the solvent molecules, i.e. water. How extra hydrogen bonds with H2O molecules could alter the DOX adhesion?
Response: Thanks for your advice. The direct binding of G and GO with DOX is affected by many factors. The main purpose of our study is to simulate the influence of different ratios of hydroxyl and epoxy groups on the adsorption of doxorubicin. The boundary of G and GO and solvent molecules will also have an impact on adsorption, which will be taken into account in subsequent calculations and experiments.
Reviewer 2 Report
The authors should answer the following questions before the paper can be published in Nanomaterials:
1) On line 128, equation 1 appears to be printed incorrectly. The equation should be proofread and corrected. The equation as printed does not include either EGS or EDOX.
2) On line 162, the phrase “structure of DOX changed” is used when it appears “orientation of DOX changed” was intended. The discussion in that paragraph involves the way DOX orients relative to the graphene sheet.
3) According to Table 1, all of the graphene oxides adsorbed DOX less strongly than pristine graphene. How can this be, when the graphene oxides have polar functional groups that should be attractive to the DOX?

Author Response
Q1: On line 128, equation 1 appears to be printed incorrectly. The equation should be proofread and corrected. The equation as printed does not include either EGS or EDOX.
Response: Due to our typesetting mistake, Equation (1) has been omitted. We have added the Equation (1) in the revised manuscript
Q2: On line 162, the phrase “structure of DOX changed” is used when it appears “orientation of DOX changed” was intended. The discussion in that paragraph involves the way DOX orients relative to the graphene sheet.
Response: Thanks for the correction. We want to express that DOX's orientation has changed. Our previous expression was not accurate enough, and we have corrected it in the manuscript.
Q3: According to Table 1, all of the graphene oxides adsorbed DOX less strongly than pristine graphene. How can this be, when the graphene oxides have polar functional groups that should be attractive to the DOX?
Response: The adsorption strength of DOX decreases when oxidized functional groups are introduced onto graphene surface. Structural and RDG analysis also confirmed that the adsorption between DOX and pristine graphene is mainly due to π -- π stacking interaction. When oxidized functional groups are introduced, the conjugated structure of graphene surface is broken. This π -- π stacking interaction is reduced, and the introduced polar functional groups form hydrogen bonds with DOX to facilitate adsorption, but not as strong as DOX with pristine graphene. However, raw graphene has high hydrophobicity, which hinders its dispersion in the blood and causes hemolysis, thus limiting its use as a drug carrier. Go has higher stability and safety, and is more suitable as a drug carrier. However, doxorubicin has weak binding ability to GO, so it is very necessary to find a method to improve the binding ability of GO and doxorubicin. Since different proportions of polar functional groups can affect the strength of hydrogen bonds between DOX and GO, our study also mainly analyzed the effect of changing the ratio of oxygen functional groups on the adsorption strength of doxorubicin. We calculated the possibility of increasing the DOX loading capacity of GO by increasing the ratio of hydroxyl to epoxy groups.
Reviewer 3 Report
The manuscript of Song et al. describes a computational study of the interaction of the anti-cancer drug doxorubicin with graphene and and its derivative graphene oxide. The goal of this study is to provide a robust characterization, from a quantum-chemical point of view, of the effects of the chemical functionalization of graphene with hydroxyl and epoxy groups on the binding energy of doxorubicin to the surface. This can guide the development of new physiologically compatible carbon-based materials acting as drug carriers for doxorubicin.
The authors use density-functional theory calculations and ab initio molecular dynamics simulations to study the binding energy of a doxorubicin molecule to graphene and a series of graphene oxide samples with different ratios of hydroxyl and epoxy groups. They also carry out extensive electron density analysis to interpret the chemical mechanism of doxorubicin-graphene/graphene oxide biding. Their results indicate that the loading capacity of graphene oxides for doxorubicin can be enhanced and optimized by varying the ratio of hydroxyl versus epoxy groups.
The study is focussed and its objectives are clear and well defined. The methods used are sound and the results of the calculations support the main conclusions of the work. The length and level of detail are appropriate. I recommend publication in Nanomaterials, but I would like the authors to address the following minor point prior to publication.
1) All calculations were carried out as if the graphene/graphene oxide surface and the doxorubicin molecule were in the gas phase. One may argue that, in physiological conditions, there could be important changes in the nature of the interactions between the drug and the surface. For instance, deportation of -OH groups could limit the ability of graphene oxide to act as an electron acceptor. Hydrogen bonding could also play an important role. Although simulating the adsorption of doxorubicin in solvent (e.g. water) using the methods adopted in this paper could be computationally too demanding, it would be nice to add a few comments to discuss how the results presented in this work could be affected by the presence of a physiological environment.
2) Although a discussion of the ability of graphene oxide to act as a drug carrier for doxorubicin in terms of binding energy is certainly a useful starting point, one should also consider that too high a binding energy could prevent release of the drug. Therefore, a more complete analysis of the ability of graphene oxide to act as a drug carrier should also take into account the mechanism of the drug release and the influence of hydroxy/epoxy ratio on both drug adsorption and desorption.
3) Although it is clear from a theoretical point of view that modifying the hydroxy/epoxy ratio can influence doxorubicin adsorption, it remains to be understood whether one can achieve desired ratios in practical conditions. A brief comment on the potential of experimental methods to synthesize graphene oxides with desired hydroxy/epoxy ratios could be useful.
Author Response
Q1: All calculations were carried out as if the graphene/graphene oxide surface and the doxorubicin molecule were in the gas phase. One may argue that, in physiological conditions, there could be important changes in the nature of the interactions between the drug and the surface. For instance, deportation of -OH groups could limit the ability of graphene oxide to act as an electron acceptor. Hydrogen bonding could also play an important role. Although simulating the adsorption of doxorubicin in solvent (e.g. water) using the methods adopted in this paper could be computationally too demanding, it would be nice to add a few comments to discuss how the results presented in this work could be affected by the presence of a physiological environment.
Response: Thank you very much for your advice. DFT can only be used to calculate a simplified model, which cannot simulate the influence of other factors in the real environment. The interaction of the adsorption complex may be influenced by water molecules in the liquid environment under physiological conditions, to which we have added a discussion in the manuscript. " Due to our calculations were performed in a vacuum. The adsorption of GO surface and doxorubicin molecules is also affected by water in the physiological environment, which may compete with DOX for the active sites on GO surface and to some extent weaken the interaction between GO and DOX. As a result, the adsorption intensity between GO and DOX in physiological environment is weaker than the result obtained by our calculation."
Q2: Although a discussion of the ability of graphene oxide to act as a drug carrier for doxorubicin in terms of binding energy is certainly a useful starting point, one should also consider that too high a binding energy could prevent release of the drug. Therefore, a more complete analysis of the ability of graphene oxide to act as a drug carrier should also take into account the mechanism of the drug release and the influence of hydroxy/epoxy ratio on both drug adsorption and desorption.
Response: Excessively increasing the binding capacity between DOX of GO will prevent drug release, which requires us to improve the binding capacity within a certain range. At present due to low bonding strength in cell culture medium of graphene oxide immediately to passive control release DOX, acidic pH stimulus can effectively make graphene oxide release DOX, normal physiological environment of pH is 7.4, about 5-5.5 cancerous tumors near, all we want to appropriate to increase the ability of a union between a graphene oxide and a DOX helps with passive targeting of our drug delivery system. The optimal range of quantitative analysis and combination ability needs further study.
Q3: Although it is clear from a theoretical point of view that modifying the hydroxy/epoxy ratio can influence doxorubicin adsorption, it remains to be understood whether one can achieve desired ratios in practical conditions. A brief comment on the potential of experimental methods to synthesize graphene oxides with desired hydroxy/epoxy ratios could be useful.
Response: Current experimental means can adjust the proportion of oxidizing functional groups of GO to a certain extent by using different chemical oxidation methods, the dosage of oxidizing agent, reaction temperature and reaction time. Therefore, we believe that it is feasible to gradually increase the hydroxyl/epoxy ratio on the surface of GO by experimental means, observe the difference of DOX loading capacity, and find the optimal experimental conditions.
Reviewer 4 Report
I found this paper very interesting and of high relevance to model the drug delivery process. The authors modelled the interactions of the title drug doxorubicine with graphene and different oxidation products of it. The results obtained with molecular dynamics and DFT were used to perform the density analysis.
My problem with the paper is that it does not provide clear picture of what has been done in a way understandable for the readers not belonging to the molecular dynamics and periodic boundary conditions community.
My list of comments in this regard:
-Figure 1 does not give insight into the structure of the doxorubicine. Particularly, the Please add a classical structure formula, so the methyloxan ring and the chiral centres could be seen.
-Lines 89-91. The whole description is somewhat confusing. The authors suggest to used the 308 carbon unit as the model of graphene but then indicate to use the periodic boundary conditions with "a vacuum plate with a thickness 90
of 35 Å was added to prevent interactions between adjacent unit cells. "
I believe the a reader needs a figure to get the insight how does look in space. I got a problem to grasp what is a dimensionality of the used model.
-Paragraphes 2.2 and 2.3. I think it may be more clear for the readers of broader community if the authors stated explicitly that they first found the possibly energy minima of the adsorbed DOX with molecular dynamics and then used it as starting point of the DFT optimisation using a "solid-state" model.
A question resulting from curiosity of this reviewer. Wouldn't a "single molecule approach" with a higher level of DFT bring a more detailed picture? I mean the application of the 308 carbon model, with frozening of the coordinates of the most of graphene carbons with a higher-tier density correlation functional and large basis set using the packages like ORCA, Gaussian or the relevant options of Quantum Express or like?
-Figure 3 and the relevant text.
I have a problem to understand the difference between the models A and B. Please add some more information.
-Equation 1, line 128. Do I miss something or is there a typing error?
Author Response
Q1: Figure 1 does not give insight into the structure of the doxorubicine. Particularly, the Please add a classical structure formula, so the methyloxan ring and the chiral centres could be seen.
Response: Thank you for your suggestion. In order to make readers better understand the structure of doxorubicin, we have added the classical structure formula of doxorubicin in Figure 1.
Q2: Lines 89-91. The whole description is somewhat confusing. The authors suggest to used the 308 carbon unit as the model of graphene but then indicate to use the periodic boundary conditions with "a vacuum plate with a thickness 90 of 35 Å was added to prevent interactions between adjacent unit cells. " I believe the a reader needs a figure to get the insight how does look in space. I got a problem to grasp what is a dimensionality of the used model.
Response: Since the calculation we used is a periodic calculation in three-dimensional space, a vacuum layer is needed in the direction perpendicular to the graphene layer to avoid the interaction between graphene layers to interfere with our calculation results. We have added the vacuum layer model in Figure 2 and re-describe the graphene modeling to help the reader understand.
Q3: Paragraphes 2.2 and 2.3. I think it may be more clear for the readers of broader community if the authors stated explicitly that they first found the possibly energy minima of the adsorbed DOX with molecular dynamics and then used it as starting point of the DFT optimisation using a "solid-state" model.
Response: Thank you for your suggestions. We have made changes to 2.2 and 2.3 according to your suggestions. "To establish a more accurate adsorption model, we found the structure of DOX adsorption by graphene at the minimum possible energy by molecular dynamics. Then use this Structure as a starting point for DFT optimization."
Q4: A question resulting from curiosity of this reviewer. Wouldn't a "single molecule approach" with a higher level of DFT bring a more detailed picture? I mean the application of the 308 carbon model, with frozening of the coordinates of the most of graphene carbons with a higher-tier density correlation functional and large basis set using the packages like ORCA, Gaussian or the relevant options of Quantum Express or like?
Response: This method can obtain a more detailed image, but I think need to using molecular dynamics simulations to optimize the rough looking for possible energy minimum points, and then the DFT structure optimization, finally according to what you have said, with a high level of functional and higher truncation can to calculate the energy and wave function of optimizing the structure of good information, etc. This will greatly save computing resources and get better results.
Q5: Figure 3 and the relevant text. I have a problem to understand the difference between the models A and B. Please add some more information.
Response: We may have misinterpreted the original narrative. Both epoxy groups are below the carbon plane, and both hydroxyl groups are above the carbon plane (Figure 3a and 3b) Model A and B is two faces of the same GO, with two hydroxyl groups on one side and two epoxy groups on the other. For model A, two hydroxyl groups are on the upper side of the graphene surface and two epoxy groups are on the lower side of the graphene layer of the graphene. For model B, two epoxy groups are on the upper side of the graphene layer and two hydroxyl groups are on the lower side of the graphene surface.
Q6: Equation 1, line 128. Do I miss something or is there a typing error?
Response: Due to our typesetting mistake, Equation (1) has been omitted. We have added the Equation (1) in the revised manuscript.
Round 2
Reviewer 1 Report
The text under review is the purely computational study of adsorption of widely used anticancer agent doxorubixin - DOX on the surface of graphene and graphene oxide. The last contains functional groups of two types hydroxyl and epoxy.
Authors applied the density functional theory on the PBE level with dispersive correction to obtain the structures with optimal total energy. They calculated adsorption energies and performed topological analysis of the non-bonded interactions between atoms.
In the result of the consideration of adsorption energies the most favorable is the unmodified graphene substrate. Next favorable is the graphene oxide with hydroxyl groups, and the least – graphene oxide with epoxide groups. Also authors found that hydrogen bonds are more important than π-π interactions in the DOX adhesion process on the GO.
The results are interesting, however, I feel that their presentation could be improved.
Q1.1. Authors considered 3 variants of GO model, in particular, C8O2(OH)2 fragments, illustrated on Figure 3. But the forth variant was not considered as proposed in study of other authors [Ghaderi et al. J. Phys. Chem. C 2010, 114 (49), 21625–21630]. However, these results obtained by Ghaderi et al. and the authors of reviewed work may disagree by a variety of reasons (positive are: better precision of modern software, richer computational resourses, completeness of the models). So that, authors should revisit those results before omitting the model. And this could be easily done as a side step of already completed calculations.
Q1.2. Please specify the relaxed energies of the structures on Figure 3 a-c. How close are they?
Q1.3. Figure 3 shows three clusters C8O2(OH)2 on each model of GO. Why 3? Why at this specific distance between them? These parameters may significantly alter the adhesion of DOX, but the only mentioning in the text is “oriented to form an equilateral triangle.”
Q2. The new phrase “found the structure of DOX adsorption by graphene at the minimum possible energy by molecular dynamics.” is confusing. MD provides trajectory, not configuration with minimum possible energy. That is obtained by “relaxation” or “optimization”. Please correct.
Q3. Please don’t use the engineering exponent symbol “E” for the powers of 10. “3E-3 Bohr” etc. Please describe the convergence tests for the selection of cutoff energy.
Q4. Please specify the value of “partial charge transfer” (Section 3.3.). This could be done by integration of delta density. If it is not possible, specify the iso-value used on Figure 9.
Q5. Page 6 “Only the atomic calculated wave function need to be entered”. Probably should be “electron density” instead of “wave functions”.
Q6. The were probably made in a hurry resulting in a poor gramatics (up to my limited knowledge of English). For example: “adsorption intensity”, “atoms of O1, O2, O3 and O4 can be found in Figure 1” etc. Please re-read and correct the text.
Author Response
Q1.1. Authors considered 3 variants of GO model, in particular, C8O2(OH)2 fragments, illustrated on Figure 3. But the forth variant was not considered as proposed in study of other authors [Ghaderi et al. J. Phys. Chem. C 2010, 114 (49), 21625–21630]. However, these results obtained by Ghaderi et al. and the authors of reviewed work may disagree by a variety of reasons (positive are: better precision of modern software, richer computational resourses, completeness of the models). So that, authors should revisit those results before omitting the model. And this could be easily done as a side step of already completed calculations.
Response:Thank you for your suggestion. We will carefully review the results referred to in the follow-up research. And more possible models will be considered in future studies.
Q1.2. Please specify the relaxed energies of the structures on Figure 3 a-c. How close are they?
Response:Thank you for your suggestion. In order to make readers more clearly understand the structural differences of a-c in Figure 3, we have added the relaxation energy of a-c structure in Figure 3 in Table 1 of the paper.
Q1.3. Figure 3 shows three clusters C8O2(OH)2 on each model of GO. Why 3? Why at this specific distance between them? These parameters may significantly alter the adhesion of DOX, but the only mentioning in the text is “oriented to form an equilateral triangle.”
Response:An equilateral triangle is formed by selecting three graphene sheets and distributing them evenly. This arrangement allows as many oxygen-containing functional groups as possible to interact with DOX. We added more explanations for the establishment of this model in the paper.
Q2. The new phrase “found the structure of DOX adsorption by graphene at the minimum possible energy by molecular dynamics.” is confusing. MD provides trajectory, not configuration with minimum possible energy. That is obtained by “relaxation” or “optimization”. Please correct.
Response:Thank you for your correction. There may be some ambiguity in our previous expression. Here, molecular dynamics simulations were used to observe the adsorption between DOX and GO. By observing the adsorption trajectory, the rough optimal adsorption structure was found, and then the optimal adsorption structure was calculated by using DFT structure optimization. We have corrected it in the article.
Q3. Please don’t use the engineering exponent symbol “E” for the powers of 10. “3E-3 Bohr” etc. Please describe the convergence tests for the selection of cutoff energy.
Response:Thanks for your reminding. We have corrected this expression in the manuscript. For the convergence test of truncation energy, we selected the GO system b in Fig. 3 to test the single point energy calculation. In the selection of 300Ry, 400 Ry, 450 Ry, 500 Ry, 550 Ry and 600 Ry, the test was carried out. It is found that when the truncation energy is greater than 500, the energy changes little while the calculation time increases greatly. Therefore, 500Ry is selected as the truncation energy.
Q4. Please specify the value of “partial charge transfer” (Section 3.3.). This could be done by integration of delta density. If it is not possible, specify the iso-value used on Figure 9.
Response:Thanks for your suggestion, we used Bader charge analysis to calculate the value of charge transfer. It has been added in the manuscript. "When DOX was adsorbed onto pristine graphene, the pristine graphene transferred 0.04 electrons to DOX. When DOX was adsorbed onto GO, 0.05 electrons were transferred from DOX to GO-O and GO-OH-O surface, respectively. And 0.07 electrons are transferred from DOX to the GO-OH surface."
Q5. Page 6 “Only the atomic calculated wave function need to be entered”. Probably should be “electron density” instead of “wave functions”.
Response:Thank you for your correction. There may be ambiguity in our previous statement. What we input is the wave function file calculated by DFT. Because we use Multiwfn Program for RDG analysis of computation, we only need to input the wave function file calculated by DFT to obtain the RDG analysis results.
Q6. The were probably made in a hurry resulting in a poor gramatics (up to my limited knowledge of English). For example: “adsorption intensity”, “atoms of O1, O2, O3 and O4 can be found in Figure 1” etc. Please re-read and correct the text.
Response:Thanks for your correction. We have re-read the manuscript and revised the inaccurate grammar.